# Phytoceramides from the Marine Sponge *Monanchora clathrata*: Structural Analysis and Cytoprotective Effects

**DOI:** 10.3390/biom13040677

**Published:** 2023-04-14

**Authors:** Elena A. Santalova, Alexandra S. Kuzmich, Ekaterina A. Chingizova, Ekaterina S. Menchinskaya, Evgeny A. Pislyagin, Pavel S. Dmitrenok

**Affiliations:** G.B. Elyakov Pacific Institute of Bioorganic Chemistry, Far Eastern Branch of the Russian Academy of Sciences, Pr. 100-let Vladivostoku 159, 690022 Vladivostok, Russia; assavina@mail.ru (A.S.K.); chingizova_ea@piboc.dvo.ru (E.A.C.); ekaterinamenchinskaya@gmail.com (E.S.M.); pislyagin@hotmail.com (E.A.P.)

**Keywords:** phytoceramides, structural analysis, NMR, ESI-MS, sponge, *Monanchora clathrata*, cytoprotective effect, neuroprotective effect

## Abstract

In our research on sphingolipids from marine invertebrates, a mixture of phytoceramides was isolated from the sponge *Monanchora clathrata* (Western Australia). Total ceramide, ceramide molecular species (obtained by RP-HPLC, high-performance liquid chromatography on reversed-phase column) and their sphingoid/fatty acid components were analyzed by NMR (nuclear magnetic resonance) spectroscopy and mass spectrometry. Sixteen new (**1b**, **3a**, **3c**, **3d**, **3f**, **3g**, **5c**, **5d**, **5f**, **5g**, **6b**–**g**) and twelve known (**2b**, **2e**, **2f**, **3b**, **3e**, **4a**–**c**, **4e**, **4f**, **5b**, **5e**) compounds were shown to contain phytosphingosine-type backbones *i*-t17:0 (**1**), *n*-t17:0 (**2**), *i*-t18:0 (**3**), *n*-t18:0 (**4**), *i*-t19:0 (**5**), or *ai*-t19:0 (**6**), *N*-acylated with saturated (2*R*)-2-hydroxy C_21_ (**a**), C_22_ (**b**), C_23_ (**c**), *i-*C_23_ (**d**), C_24_ (**e**), C_25_ (**f**), or C_26_ (**g**) acids. The used combination of the instrumental and chemical methods permitted the more detailed investigation of the sponge ceramides than previously reported. It was found that the cytotoxic effect of crambescidin 359 (alkaloid from *M. clathrata*) and cisplatin decreased after pre-incubation of MDA-MB-231 and HL-60 cells with the investigated phytoceramides. In an in vitro paraquat model of Parkinson’s disease, the phytoceramides decreased the neurodegenerative effect and ROS (reactive oxygen species) formation induced by paraquat in neuroblastoma cells. In general, the preliminary treatment (for 24 or 48 h) of the cells with the phytoceramides of *M. clathrata* was necessary for their cytoprotective functions, otherwise the additive damaging effect of these sphingolipids and cytotoxic compounds (crambescidin 359, cisplatin or paraquat) was observed.

## 1. Introduction

Ceramide is a complex lipid, which consists of a sphingoid base attached to a fatty acid via an amide bond. Ceramides are present in membranes in small amounts only, but they serve as central mediators, regulating many fundamental cellular responses [1,2,3]. These sphingolipids trigger a number of tumor suppressive and anti-proliferative cellular programs, for example, apoptosis, autophagy, senescence, and necroptosis [4,5]. Ceramide-dependent effects may hold implications for the progression of many diseases including cancer, diabetes, atherosclerosis, Alzheimer’s and Parkinson’s diseases [1].

Ceramides vary appreciably in the compositions of both long-chain alkyl (sphingoid and fatty acid) components, depending on their biological origins. In particular, phytoceramides, consisting of phytosphingosine-type backbones *N*-acylated with 2-hydroxy fatty acids, are common for marine sponges [6]. In our research on sphingolipids from marine invertebrates, phytoceramides, mainly containing *iso*-methyl-branched chains, were found in the sponge *Monanchora clathrata*, collected in Australian waters.

The phytoceramides of the sponge *M. clathrata* have been the subjects of a study reporting the structures of four new compounds [7]. However, since the MS/MS (tandem mass spectrometry) technique was not used in the previous study, it remained possible that the ceramide composition of *M. clathrata* was more complex than that reported. The present work was undertaken in order to provide more detailed information on the ceramide composition of *M. clathrata* by applying a combination of chemical and instrumental methods including RP-HPLC (high-performance liquid chromatography on reversed-phase column), NMR (nuclear magnetic resonance), ESI-MS (electrospray ionization mass spectrometry), ESI-MS/MS, and GC-MS (gas chromatography-mass spectrometry). The phytoceramides of *M. clathrata* were analyzed as constituents of multi-component RP-HPLC fractions, and the structures of 16 new (**1b**, **3a**, **3c**, **3d**, **3f**, **3g**, **5c**, **5d**, **5f**, **5g**, **6b**–**g**) and 12 known (**2b**, **2e**, **2f**, **3b**, **3e**, **4a**–**c**, **4e**, **4f**, **5b**, **5e**) compounds were elucidated (Figure 1).

In the previous study of the phytoceramides of *M. clathrata*, these compounds were shown to be cytotoxic to MES-SA, MCF-7, and HK-2 cell lines [7]. The results from other studies suggest that phytoceramides may help to prevent neurodegeneration in vitro and in vivo [9,10]. The neuroprotective effect of phytoceramides is consistent with a possible therapeutic role of these compounds in managing cognitive impairment, associated, in the first turn, with Alzheimer’s disease. We aimed to investigate mainly the cytoprotective effects of the phytoceramides from *M. clathrata*, in particular their effect on paraquat-induced neurotoxicity using in vitro model of Parkinson’s disease. The neuroprotective activity of phytoceramides in Parkinson’s disease has not been studied, although this pathology is the second most common neurodegenerative disorder behind Alzheimer’s disease [11]. Alkaloid crambescidin 359 (**7**), previously isolated from the sample of *M. clathrata* analyzed here [12], was also used in our bioassays. The cytotoxic and/or cytoprotective effects of total ceramide, crambescidin 359 and their combinations were tested on MDA-MB-231 (breast adenocarcinoma) and HL-60 (leukemia) cells. The cytotoxic effect of cisplatin in combinations with the phytoceramides of *M. clathrata* was also tested on these cells. Cisplatin is employed for the treatment of various tumors, and there is an urgent need for therapeutic agents reducing cisplatin-induced toxicity. Based on the results of the present study, it is concluded that, depending on the time of cell pre-treatment by phytoceramides, these sphingolipids can increase or decrease the cytotoxicity of crambescidin 359, cisplatin or paraquat.

## 2. Materials and Methods

### 2.1. General Procedures

^1^H-, ^13^C-NMR, ^1^H,^1^H-COSY and HSQC spectra (in C_5_D_5_N or CDCl_3_) were recorded on Bruker Avance III HD 500 and Bruker Avance III 700 spectrometers (Bruker BioSpin, Bremen, Germany) at 125 MHz (^13^C), 500 (^1^H), and 700 (^1^H) MHz. A Bruker Impact II Q-TOF mass spectrometer (Bruker Daltonik GmbH, Bremen, Germany) equipped with ESI ionization source was employed to record MS and MS/MS spectra. The operating parameters for ESI-MS were the following: a capillary voltage of 4.0 kV, nebulization with nitrogen at 0.4 bar, and a dry gas flow of 4 L/min at a temperature of 200 °C. GC analyses were performed on an Agilent 6850 Series GC System chromatograph (Agilent Technologies, Santa Clara, CA, USA) equipped with an DB-1 (J&W Scientific, Folsom, CA, USA) capillary column (30 m × 0.32 mm), the carrier gas was helium (flow rate 1.7 mL/min), and the detector temperature was 300 °C. GC-MS analyses were carried out on a Hewlett-Packard HP6890 GC System (Hewlett-Packard Company, Palo Alto, CA, USA) with an HP-5MS (J&W Scientific, Folsom, CA, USA) capillary column (30.0 m × 0.25 mm), helium as the carrier gas, and 70 eV ionizing potential. The GC and GC-MS analyses of fatty acid esters and peracetylated sphingoid bases were performed using the injector temperature of 270 °C and the temperature program 100 °C (1 min) − 10 °C/min − 280 °C (30 min). Optical rotation was measured on a Perkin–Elmer polarimeter, model 343 (Perkin-Elmer GmbH, Überlingen, Germany). Column chromatography was performed using silica gel (50/100 μm, Sorbpolimer, Krasnodar, Russia). RP-HPLC separations were performed using a Du Pont Series 8800 Instrument (DuPont, Wilmington, DE, USA) with a RIDK-102 refractometer (Laboratorni pristroje, Praha, Czechoslovakia). An Agilent ZORBAX Eclipse XDB-C8 column (4 × 150 mm; Agilent Technologies, Santa Clara, CA, USA) was used for the HPLC.

### 2.2. Animal Material

The sponge *Monanchora clathrata* (phylum Porifera, class Demospongiae, order Poecilosclerida, family Crambeidae) was collected by divers from 7 m depth near the Western Australian coast (26°09.8/S, 113°12.8/E) during a cruise onboard the r/v “Akademik Oparin” in September, 1987. The species was identified by V.B. Krasokhin (G.B. Elyakov Pacific Institute of Bioorganic Chemistry, Russia).

### 2.3. Extraction and Isolation

The collected sponge was cut, lyophilized, stored at −15 °C, and then extracted with EtOH at room temperature. After evaporation in vacuo, EtOH extract (4.6 g) was partitioned between H_2_O (200 mL) and *n-*BuOH (150 mL). The dry *n-*BuOH extract was separated to give total ceramide and ceramide HPLC fractions (Figure 1). Total ceramide: amorphous solid; ^1^H- and ^13^C-NMR (500 MHz, C_5_D_5_N): see Table 1; high resolution ESI-MS in negative ion mode (HR-(–)ESI-MS): 640.5886 ([M − H]^−^ of **1b**, **2b**, **3a**, and **4a**, C_39_H_78_NO_5_^−^; calc. 640.5885); 654.6045 ([M − H]^−^ of **3b** and **4b**, C_40_H_80_NO_5_^−^; calc. 654.6042); 668.6196 ([M − H]^−^ of **2e**, **3c**, **3d**, **4d**, **5b**, and **6b**, C_41_H_82_NO_5_^−^; calc. 668.6198); 682.6354 ([M − H]^−^ of **2f**, **3e**, **4e**, **5c**, **5d**, **6c**, and **6d**, C_42_H_84_NO_5_^−^; calc. 682.6355); 696.6510 ([M − H]^−^ of **3f**, **4f**, **5e**, and **6e**, C_43_H_86_NO_5_^−^; calc. 696.6511); 710.6665 ([M − H]^−^ of **3g**, **5f**, and **6f**, C_44_H_88_NO_5_^−^; calc. 710.6668); and 724.6828 ([M − H]^−^ of **5g** and **6g**, C_45_H_90_NO_5_^−^; calc. 724.6824).

HCl-Catalyzed hydrolysis was performed according to the procedure of Aveldaño and Horrocks [13]. A part (7.0 mg) of the total ceramide was hydrolyzed in MeCN (0.45 mL) and 5 N HCl (0.05 mL) for 4 h at 70 °C. The reaction mixture was concentrated in vacuo and acetylated with Ac_2_O in pyridine (1:1, *v*/*v*, 0.2 mL, overnight). The acetylated material was separated by chromatography on silica gel (column: 4.5 cm × 1.5 cm) using hexane/ethylacetate (5:1 → 2:1 → 1:1, *v*/*v*) systems, then CHCl_3_/EtOH (1:1) system. Elution with hexane/ethylacetate (2:1, *v*/*v*, 60 mL) gave tetracetates of sphingoid bases (3.5 mg): [α]^22^_D_ = +25.5 (*c* = 0.2, CHCl_3_); ^1^H-NMR (700 MHz, CDCl_3_): 5.96 (d, *J* = 9.3 Hz, NH), 5.10 (dd, *J* = 3.1, 8.2 Hz, CH-3), 4.94 (dt, *J* = 3.1, 10.1 Hz, CH-4), 4.47 (m, CH-2), 4.28 (dd, *J* = 4.9, 11.6 Hz, CH-1b), 4.005 (dd, *J* = 3.1, 11.6 Hz, CH-1a), 2.075 (s, −OCOC**H**_3_ at C-3), 2.045 (s, −OCOC**H**_3_ at C-4 and at C-1), 2.02 (s, −NHCOC**H**_3_), 1.63 (m, CH_2_-5), 1.51 (sep, *J* = 6.6 Hz, –C**H**(CH_3_)_2_ of *iso*-methyl-branched components), 1.40 − 1.20 (m, CH_2_-pool), 1.15 (m, –C**H_2_**CH(CH_3_)_2_ of *iso*-methyl-branched components), 0.88 (t, *J* = 7.0 Hz, terminal –CH_3_ of minor *normal*-chain components), 0.86 (d, *J* = 6.6 Hz, terminal −CH_3_ of *iso*-methyl-branched components), 0.85 (t, *J* = 7.0 Hz, −CH(CH_3_)CH_2_C**H**_3_ of minor *anteiso*-methyl-branched components), 0.84 (d, *J* = 6.0 Hz, −CH(C**H**_3_)CH_2_CH_3_ of minor *anteiso*-methyl-branched components); GC-MS: see Section 3.2 below. Elution with CHCl_3_/EtOH (1:1, 20 mL) gave 2-acetyloxy fatty acids. These acids were treated with 2% H_2_SO_4_ in (2*R*)-octan-2-ol or (2*S*)-octan-2-ol (0.2 mL) for 4 h at 75 °C in a capped vial [14] to obtain (2*R*)- and (2*S*)-oct-2-yl esters of 2-hydroxy fatty acids.

The RP-HPLC purification of the tetracetylated sphingoid bases (3.5 mg) on an Agilent ZORBAX Eclipse XDB-C8 column (70% EtOH) yielded subfractions I (0.1 mg: *n*-t17:0, 94.4%; *i*-t17:0, 5.6%), II (1.9 mg: *i*-t18:0, 98.2%; *n*-t18:0, 1.8%), III (0.3 mg: *i*-t18:0, 50.9%; *n*-t18:0, 32.8%; *ai*-t19:0, 16.3%), and IV (0.6 mg: *i*-t19:0, 26.0%; *ai*-t19:0, 74.0%).

Methanolysis of ceramides (73.6 mg) in MeOH (2.0 mL) and HCl (0.4 mL) for 4 h at 90 °C gave methyl esters of 2-hydroxy acids (19.2 mg): [α]^22^_D_ = −5.2 (*c* = 0.2, CHCl_3_); ^1^H-NMR (CDCl_3_, 300 MHz): 4.185 (m, H-2), 3.79 (s, −OCH_3_), 2.66 (d, *J* = 5.8 Hz, −OH), 1.78 (m, H-3b), 1.64 (m, H-3a), 1.50−1.20 (m, CH_2_-pool), 0.88 (t, *J* = 6.9 Hz, terminal –CH_3_ of dominant straght-chain compounds) and 0.86 (d, *J* = 6.6 Hz, terminal –CH_3_ of minor *iso*-methyl-branched compound); GC-MS: see Section 3.2 below. The methyl esters of 2-hydroxy acids were converted to DMOX and pyrrolidine derivatives, as described earlier [15].

The acid hydrolysis of ceramide molecular species (Fractions I–VII), obtained by RP-HPLC, was carried out under the same conditions as for total ceramide. Water (0.2 mL) was added to hydrolysate, and MeCN–H_2_O layer was extracted with hexane (5 × 0.5 mL). Hexane extract was dried in vacuo, acetylated with Ac_2_O in pyridine (1:1, *v*/*v*, 0.2 mL, overnight) and ethylated with *N*-nitroso-*N*-ethylurea. The resulting tetraacetates of sphingoid bases and ethyl esters of 2-acetyloxy acids were analyzed by GC-MS. The ethyl esters dominated in the mixture, and their signals and some signals of tetraacetylated sphingoid bases overlapped in GC profiles. To obtain a pure subfraction of the acetates of sphingoid bases, the mixture was separated by chromatography on silica gel (column: 3.0 cm × 1.5 cm) using hexane/ethylacetate (5:1 → 2:1, *v*/*v*). Elution with hexane/ethylacetate, 2:1 (60 mL) gave tetraacetates of sphingoid bases that were again analyzed by GC-MS.

### 2.4. Cytotoxic and Cytoprotective Activities

#### 2.4.1. Reagents

Dulbecco’s Modified Eagle Medium (DMEM), Roswell Park Memorial Institute medium (RPMI 1640), Minimum Essential Medium (MEM), 1% penicillin-streptomycin solution and 10% fetal bovine serum (FBS) were purchased from *Biolot* (Moscow or St. Petersburg, Russia), penicillin-streptomycin solution, 10 μg/mL, and dimethyl sulfoxide (DMSO) from Sigma-Aldrich, 3-(4,5-dimethylthiazol-2-yl)-5-(3-carboxymethoxyphenyl)-2-(4-sulfophenyl)-2H-tetrazolium (MTS reagent) from Promega (Madison, WI, USA), 3-(4,5-dimethylthiazol-2-yl)-2,5-diphenyltetrazolium bromide (MTT) from Sigma-Aldrich (St. Louis, MO, USA), 2,7-dichlorodihydrofluorescein diacetate solution (10.0 µM H2DCFDA) from Molecular Probes (Eugene, OR, USA), and cisplatin from *VeroPharm* (Moscow, Russia).

#### 2.4.2. Cell Lines and Culture Conditions

Human breast cancer MDA-MB-231 cells (ATCC^®^ HTB-26™), acute promyelocytic leukemia HL-60 cells (ATCC^®^ CCL-240™), neuroblastoma SH-SY5Y cells (CRL-2266™) and murine neuroblastoma Neuro-2a cells (CCL-131™) were obtained from the American Type Culture Collection (Manassas, VA, USA). The MDA-MB-231 and HL-60 cells were cultured in complete DMEM/10% FBS and RPMI 1640/10% FBS, respectively, containing 1% of penicillin-streptomycin solution. The SH-SY5Y cells were cultured in MEM containing 10% FBS and 1% penicillin-streptomycin solution. The Neuro-2a cells were cultured in DMEM containing 10% FBS and 1% penicillin-streptomycin solution. The cell cultures were incubated at 37 °C in a humidified atmosphere containing 5% (*v*/*v*) CO_2_.

#### 2.4.3. Cell Viability Assay for MDA-MB-231 and HL-60 Cells

The effect of compounds on cell viability was evaluated using reduction of MTS into formazan product. The cells were cultured in 96-well plates (5000 cells/well) with the corresponding medium (100 µL/well, containing 10% FBS) for 12 h. The cells were treated with compounds at various concentrations (0–200 µM in DMSO) for 24 or 48 h. Then, MTS reagent (20 µL) was added into each well, and, after 4 h, MTS reduction was measured spectrophotometrically at 492 and 690 nm as background, using a Power Wave XS microplate reader (BioTek, Winooski, VT, USA). Cisplatin was used as a positive control.

#### 2.4.4. Paraquat Induced In Vitro Model of Neurotoxicity

Crambescidin 359 and ceramide were dissolved in DMSO to obtain stock solutions (10 mM concentrations). Cells SH-SY5Y and Neuro-2a (1 × 10^4^ cells/well) were supplemented with the test compounds (20 µL in PBS) at final concentrations of 0.1, 1.0, or 10.0 µM, and preincubated for 1, 24 or 48 h. Then, the cells were treated with 1.5 mM paraquat. The cells incubated without paraquat were used as a positive control, and the cells incubated with this inducer were used as a negative control. Cell viability was measured after 24 h. For this, the medium with tested substances was changed by 100 μL of fresh culture medium containing 10 μL of MTT solution (5 mg/mL). After that, the microplate was incubated for an additional 4 h. Then, 100 µL of SDS-HCl solution (1 g SDS/10 mL dH_2_O/17 µL 6N HCl) was added and incubated for 18 h. Dye absorbance was measured using a plate format spectrophotometer at a wavelength of 570 nm (Thermo Scientific, Waltham, MA, USA). All the experiments were carried out three times. The toxic activity of paraquat was expressed as a percentage of control (untreated) cells.

#### 2.4.5. Analysis of ROS (Reactive Oxygen Species) Level

Neuro-2a cells were transferred to a microplate for adhesion for 24 h, then incubated with ceramide or crambescidin 359 at various concentrations for 1h. To increase ROS in cells, paraquat was added at a concentration of 1mM per 3h. In order to study ROS formation, 20 µL of H2DCFDA solution (100 µM concentration) were added to each well (to a final concentration of 10.0 µM), and the microplate with cells was incubated for an additional 10 min at 37 °C. The intensity of fluorescence was measured using PHERAstar FS plate reader (BMG Labtech, Ortenberg, Germany) at λex = 485 nm and λem = 520 nm.

#### 2.4.6. Statistical Analysis

All experiments were carried out in three or more independent experiments. Plot data were presented as mean ± standard deviation (SD). Student’s *t*-test was performed using SigmaPlot 14.0 (Systat Software Inc., San Jose, CA, USA) to determine statistical significance.

## 3. Results

### 3.1. Analysis of the Total Ceramide of M. clathrata

The ^1^H- and ^13^C-NMR spectra (C_5_D_5_N) of the total ceramide, isolated from *M. clathrata*, showed the signals of saturated phytosphingosine-type backbones, *N*-acylated with saturated 2-hydroxy acids (Table 1, Appendix A). In the ^1^H,^1^H-COSY spectrum, the signal of –N**H**–CO– (*δ*_H_ 8.58, d, *J* = 9.0 Hz), characteristic of the sphingoid base moieties of ceramides, displayed a cross-peak with CH-2 (*δ*_H_ 5.12, m). The ^1^H,^1^H-COSY diagram also indicated that several protons, from CH_2_-1 (*δ*_H_ 4.44, dd; 4.53, dd) to CH_2_-6 (*δ*_H_ 1.48, m; 1.72, m), formed a linear spin system of phytosphingosine-type moieties. Another spin system consisted of CH-2^/^ (*δ*_H_ 4.635, dd), CH_2_-3^/^ (*δ*_H_ 2.06, m; 2.25, m), and CH_2_-4^/^ (*δ*_H_ 1.77, m) protons of the 2-hydroxy acyls. Accordingly, the signals of one CH_2_ (*δ*_C_ 61.9, CH_2_-1) and three CH (*δ*_C_ 72.3, CH_2_-2^/^; 72.9, CH_2_-4; 76.7, CH_2_-3), bearing –OH groups, and the signals of –C=O (*δ*_C_ 175.1, C-1^/^) and CH (*δ*_C_ 52.9, CH-2), linked to –NH–, were observed in the ^13^C-NMR spectrum of the investigated phytoceramides. The ^1^H- and ^13^C- NMR spectra of these compounds also showed signals of long hydrocarbon chains with terminal methyl groups belonging to *normal*-chain, *iso*- and minor *anteiso*-methyl branched constituents (Table 1). These spectra resembled the corresponding spectra of monanchoramides A–D, previously isolated from Philippine sample of the sponge *M. clathrata* [7]. Monanchoramides A–D were reported to contain *iso*-methyl-branched saturated phytosphingosine-type C_18_–C_21_ backbones, *N*-acylated with saturated *normal*-chain (2*R*)-2-hydroxy C_22_ acid.

The molecular formulae of the phytoceramides from *M. clathrata* were determined by HR-ESI-MS (high resolution ESI-MS) analyses in negative ion mode (Appendix A). The HR-(−)ESI-MS analysis of total ceramide resulted in a series of peaks, representing seven homologous [M − H]^−^ ions. ESI-MS/MS experiments showed that these molecular species contained C_17_–C_19_ sphingoid bases, linked to C_21_–C_26_ acids (Table 2).

### 3.2. Analysis of the 2-Hydroxy Fatty Acids and Sphingoid Bases Obtained from the Total Ceramide of M. clathrata

We applied methanolysis and hydrolysis for the chemical degradation of total ceramide. Methanolysis procedure was used to prepare fatty acid methyl esters. Hydrolysis was mainly used to release sphingoid bases because sphingoid bases could be harmed by vigorous conditions of methanolysis [16].

The ^1^H-NMR spectrum (CDCl_3_) of the methyl esters of 2-hydroxy acids displayed characteristic signals at δ_H_ 4.185 (m, H-2), 3.79 (s, −OCH_3_), and 2.66 (d, *J* = 5.8 Hz, −OH at CH-2). This spectrum also showed signals of methyl groups belonging to dominant *normal*-chain (δ_H_ 0.88, t, *J* = 6.9 Hz) and minor *iso*-methyl-branched (δ_H_ 0.86, d, *J* = 6.6 Hz) acyls. In GC-MS analysis, the methyl esters of 2-hydroxy acids fragmented to give diagnostic [MeOCOCH_2_OH]^+^ (*m*/*z* 90) and [M−59]^+^ ions [17]. The GC-MS analysis of these fatty acid esters showed the presence of seven compounds, two of which were isomeric (Table 3). Then, the methyl esters of 2-hydroxy acids were converted to 4,4-dimethyloxazoline and pyrrolidine derivatives, the mass spectra of which were used to clarify the structures of six 2-hydroxy *normal*-chain C_21_–C_26_ acids and one *iso*-methyl-branched C_23_ acid [15]. (2*R*)-configurations of 2-hydroxy fatty acids, liberated by hydrolysis of phytoceramides, were determined by GC-MS analyses of the (2*R*)- and (2*S*)-oct-2-yl esters of these acids. Each (2*R*)-oct-2-yl ester eluted before the diastereomeric (2*S*)-oct-2-yl ester, as described for the elution order of the (2*R*)/(2*S*)-oct-2-yl esters of standard (2*R*)-2-hydroxy fatty acids [14]. The optical rotation value ([α]_D_^22^ = −5.2, CHCl_3_) of the methyl esters of 2-hydroxy acids, obtained from the total ceramide of *M. clathrata*, also indicated their (2*R*)-configurations [18,19].

The tetraacetylated sphingoid bases (Table 4), obtained in the result of hydrolysis of the total ceramide followed by acetylation, were separated by RP-HPLC and analyzed by GC-MS and ^1^H-NMR spectroscopy. GC-MS data allowed for the identification of the following six components in the resulting HPLC fractions: major *n*-t17:0, *i*-t18:0, *i*-t19:0, and *ai*-t19:0 acetates and minor *i*-t17:0 and *n*-t18:0 acetates. The ^1^H-NMR spectra (CDCl_3_) of all the fractions contained resonances at *δ*_H_ 5.96 (d, *J* = 9.3 Hz, NH), 5.10 (dd, *J* = 3.1, 8.2 Hz, CH-3), 4.94 (dt, *J* = 3.1, 10.1 Hz, CH-4), 4.47 (m, CH-2), 4.28 (dd, *J* = 4.9, 11.6 Hz, CH-1b), and 4.005 (dd, *J* = 3.1, 11.6 Hz, CH-1a), typical of tetraacetylated phytosphingosine-type compounds [20]. The ^1^H-NMR chemical shifts of CH_2_-1–CH-4 and optical rotation value ([α]_D_^22^ = +25.5, CHCl_3_) of the mixture of the tetraacetylated sphingoid bases indicated their (2*S*,3*S*,4*R*)-configurations [20]. The ^1^H-NMR spectra of these compounds also showed signals of the methyl groups of major *iso*-methyl-branched (*δ*_H_ 0.86, *d*, *J* = 6.6 Hz), minor *anteiso*-methyl-branched (*δ*_H_ 0.84, *d*, *J* = 6.2 Hz; *δ*_H_ 0.855, *t*, *J* = 7.2 Hz), and minor *normal*-chain (*δ*_H_ 0.88, *t*, *J* = 7.0 Hz) forms. The mass spectra of the acetates of sphingoid bases were also used for distinguishing between their terminal structures because the differences in the relative abundances of [M − AcOH − CH_3_]^+^ and [M − AcOH − CH_2_CH_3_]^+^ ions reflected the position of methyl branching in *iso*- and *anteiso*- forms [21]. In the mass spectrum of tetraacetylated *i*-t19:0, the gap of 28 amu between the peaks at *m/z* 396 [M − AcOH − CH(CH_3_)_2_]^+^ and 424 [M − AcOH − CH_3_]^+^ reflected cleavage on either side of the carbon, bearing a methyl group (–CH_2_–CH_2_-/-CH(CH_3_)-/-CH_3_). In the mass spectrum of tetraacetylated *ai*-t19:0, the significant peaks at *m/z* 410 [M − AcOH − CH_2_CH_3_]^+^ and 382 [M − AcOH − CH(CH_3_)CH_2_CH_3_]^+^ and the small peak at *m/z* 396 were characteristic (according to cleavage –CH_2_-/-CH(CH_3_)-/-CH_2_–CH_3_).

### 3.3. Structure Elucidation of the Phytoceramides of M. clathrata

The total ceramide of *M. clathrata* was separated into seven molecular species by RP-HPLC (Fractions I–VII, Table 5).

Fraction I (Table 5) was shown to contain isomeric compounds **1b** (3.3%), **2b** (30.7%), **3a** (61.3%), and **4a** (4.7%). Their molecular formula C_39_H_79_NO_5_ was established by HR-(−)ESI-MS analysis. To establish the chain length of sphingoid base and 2-hydroxy acid moieties in the phytoceramides, (−)ESI-MS/MS experiments were applied, as exemplified with the MS/MS analysis of two isomers **2b** and **3a** (Figure 2 and Appendix A). Thus, the ceramides of Fraction I were found to contain saturated C_17_ and C_18_ sphingoid bases, *N*-acylated with saturated 2-hydroxy C_22_ and C_21_ fatty acids, respectively (isomeric C_17_/C_22_ and C_18_/C_21_ structures). Hydrolysis of Fraction I gave the following two *normal*-chain (2*R*)-2-hydroxy fatty acids, identified by GC-MS: (2*R*)-2-hydroxydocosanoic (from **1b** and **2b**) and (2*R*)-2-hydroxyheneicosanoic (from **3a** and **4a**). Additionally, GC-MS analysis allowed for the identification of four phytosphingosine-type compounds in hydrolysate including (2*S*,3*S*,4*R*)-2-amino-15-methyl-hexadecane-1,3,4-triol (*i*-t17:0, from **1b**), (2*S*,3*S*,4*R*)-2-amino-heptadecane-1,3,4-triol (*n*-t17:0, from **2b**), (2*S*,3*S*,4*R*)-2-amino-16-methyl-heptadecane-1,3,4-triol (*i*-t18:0, from **3a**), and (2*S*,3*S*,4*R*)-2-amino-octadecane-1,3,4-triol (*n*-t18:0, from **4a**). The structures of fatty acids and sphingoid bases were determined based on the data obtained for the “building blocks” of the total ceramide (Section 3.2). The sphingoid bases and 2-hydroxy acids were connected according to the (−)ESI-MS/MS data, and, as a result, the structures of two new (**1b** and **3a**) and two known (**2b** and **4a**) ceramides were elucidated.

Analogously, other 24 phytoceramides were analyzed as constituents of Fractions II–VII (Table 5, Appendix A).

### 3.4. Cytoprotective and/or Cytotoxic Effects of the Phytoceramides and Crambescidin 359 from M. clathrata

The cytotoxic activity of the total ceramide of *M. clathrata* against MDA-MB-231 and HL-60 cells was extremely weak (IC_50_ values ≥ 200 μM, incubation for 48 h; Appendix A). Crambescidin 359 (**7**) exhibited weak cytotoxic activity against MDA-MB-231 (IC_50_ 130 μM) and HL-60 (IC_50_ 73 μM) cells (incubation for 24 h; Appendix A). Combinations of crambescidin 359 and total ceramide were slightly more cytotoxic for MDA-MB-231 cells (Figure 3a) and significantly inhibited the viability of HL-60 cells (Figure 3b).

The cytotoxic effect of alkaloid **7** on MDA-MB-231 and HL-60 cells decreased after the pre-incubation of these cells with the phytoceramides from *M. clathrata* (Figure 4). In particular, the pre-treatment of MDA-MB-231 cells with total ceramide (50 μM concentration) for 24 h prevented cell death induced by crambescidin 359 at the concentration of IC_50_ (Figure 4a).

Combinations of cisplatin and total ceramide were slightly more cytotoxic for MDA-MB-231 cells (Figure 5a) than only cisplatin. However, these combinations insignificantly increased the survival of HL-60 cells (Figure 5b). After 24 h pre-incubation of MDA-MB-231 and HL-60 cells with total ceramide (especially, at a concentration of 50 μM), the cytotoxic effect of cisplatin was noticeably reduced (Figure 5c,d).

Total ceramide at the concentrations of 1 and 10 μM significantly reduced ROS formation (by 29.8 ± 6.7% and 32.5 ± 3.2%, respectively) in neuroblastoma cells exposed to paraquat (Figure 6). In this experiment, phytoceramides in non-cytotoxic concentrations were incubated with the cells for 1 h before paraquat was introduced into the culture. After analogous short-time pre-treatment (1 h) of the neuroblastoma cells with total ceramide, the neurotoxicity of paraquat unexpectedly increased (Figure 7). In contrast, after a longer pre-treatment (24 or 48 h) of the cells with phytoceramides, the neurotoxic activity of paraquat decreased (Figure 8). For example, total ceramide at the concentrations of 0.1 and 1.0 μM increased the viability of Neuro 2a cells by 28.7 ± 6.3 and 23.1 ± 1.2%, respectively, compared to the control cells exposed to only paraquat (Figure 8a). Furthermore, the pre-treatment (for 48 h) of SH-SY5Y cells with total ceramide at the concentration of 10 μM prevented cell death induced by paraquat (Figure 8b).

Crambescidin 359 reduced ROS formation (maximum 17.9 ± 4.6%, 10 μM concentration) in Neuro 2a cells exposed to paraquat (Figure 5) and exhibited neuroprotective effects after 1, 24, and 48 h pre-incubation with these cells (Appendix A). However, we did not consider crambescidin 359 a promising antiparkinsonic agent because this alkaloid showed inhibitory activities in electrophysiology experiments on human α7 nicotinic acetylcholine receptors [23]. The stimulation (not inhibition) of these receptors, promoting cognitive functions, may be a perspective strategy for the treatment of Parkinson‘s disease [24].

## 4. Discussion

We conducted the structural analysis of ceramides using sensitive and specific analytical tools to detect ceramide species, differentiate between them and to elucidate individual structures. The used combination of the instrumental and chemical methods permitted the more detailed investigation of the sponge ceramides than previously reported [7]. As a result, 16 new (**1b**, **3a, 3c**, **3d**, **3f**, **3g**, **5c**, **5d**, **5f**, **5g**, **6b**–**g**) and 12 known (**2b**, **2e**, **2f, 3b**, **3e**, **4a**–**c**, **4e**, **4f**, **5b**, **5e**) compounds were found in the complex mixture of the phytoceramides from *M. clathrata*. These compounds contain phytosphingosine-type backbones *i*-t17:0 (**1**), *n*-t17:0 (**2**), *i*-t18:0 (**3**), *n*-t18:0 (**4**), *i*-t19:0 (**5**), or *ai*-t19:0 (**6**), *N*-acylated with saturated (2*R*)-2-hydroxy C_21_ (**a**), C_22_ (**b**), C_23_ (**c**), *i-*C_23_ (**d**), C_24_ (**e**), C_25_ (**f**), or C_26_ (**g**) acids. Phytoceramides **3b**, **3e**, and **3f**, consisting of *i*-t18:0 backbone and straight-chain C_22_, C_24_, and C_25_ acyls, respectively, are the most abundant in the ceramide mixture. The most remarkable features of this mixture may be considered the large amounts of constituents with methyl-branched sphingoid base moieties (*iso*-forms: 76.2%, *anteiso*-forms: 16.6%) and the extremely low amounts of components, containing common unbranched sphingoid bases (4%). Only one methyl-branched acyl (1.9%, *iso*-methyl-branched 23:0) was detected among the fatty acid residues of the investigated phytoceramides.

Phytoceramides **2b**, **2e**, **2f, 4a**–**c**, **4e**, **4f** with unbranched backbones may be found in many organisms, including terrestrial mushrooms, plants, and animals [25]. However, phytoceramides with methyl-branched backbones **1**, **3**, **5**, and **6** are far less common. The compounds of this group were mainly obtained from marine organisms including sponges ([7]: *i*-t18:0 and *i*-t19:0; [26]: *i*-t17:0, *i*-t18:0, and *i*-t19:0; [27]: *i*-t17:0; [28]: *i*-t19:0; [29]: *i*-t18:0; [30]: *i*-t19:0; [31]: *i*-t19:0 and *ai*-t19:0), starfish ([32]: *ai*-t19:0), and sea grass ([33]: *i*-t17:0 and *i*-t19:0). All the unknown phytoceramides found in the present study are also characterized by these *iso*- or *anteiso*-methyl-branched backbones. Among the new variants of *N*-acylation of the above-mentioned marine sphingoid bases, compounds **3d**, **5d**, and **6d** from *M. clathrata* have methyl branching at both the sphingoid and fatty acid (*i*-C_23_) chains.

Compounds **3b** (monanchoramide B) and **5b** (monanchoramide C) have previously been found in the Philippine sample of *M. clathrata*, along with monanchoramides A (*i*-t20:0/(2*R*)-2-OH-22:0) and D (*i*-t21:0/(2*R*)-2-OH-22:0) [7], which were not detected in the Australian sample of *M. clathrata* studied here. However, although monanchoramides do not contain *anteiso*-methyl-branched chains, the signals of unidentified minor *anteiso*-forms were found in their ^13^C-NMR spectra ([7]: Appendix A). Apparently, the differences in the ceramide compositions of the Philippine and Australian samples of *M. clathrata* may be connected with geographical and seasonal influences on their fatty acid compositions. Fatty acids are known to serve as precursors in ceramide biosynthesis, and the sensitivity of the fatty acid profiles of sponges to environmentally induced (seasonal and geographical) variations was noted earlier [34].

The sponges of the family Crambeidae and related species were shown to contain several polycyclic guanidine alkaloids including crambescidins ([35] and references cited therein). Co-occurrence of crambescidin-type alkaloids and phytoceramides was found in three crambeids, including *M. clathrata* studied here. In particular, monanchoramides A–D and alkaloids of the crambescidin group were isolated from the Philippine sample of *M. clathrata* [7,36]. In addition, ceramide *i*-t17:0/2-OH-24:0 and crambescidins were obtained from the related sponge *Crambe crambe* [27]. In our study, the pretreatment of human tumor-derived cells with the phytoceramides from *M. clathrata* decreased cell death induced by crambescidin 359 (**7**). We suggest that, similarly, these phytoceramides may help to protect sponge cells against injury, caused by their own cytotoxic crambescidin(s). Most of the crambescidin alkaloids exhibited a wide range of biological activities (including potent cytotoxicity), but little information regarding the true interaction of their polycyclic core with biological targets is known [35]. Therefore, a possible relationship between cytoprotective phytoceramides and cytotoxic crambescidins needs further investigation.

Our work showed a decrease of the cytotoxic effect of cisplatin on MDA-MB-231 and HL-60 cells after their pre-incubation with the phytoceramides of *M. clathrata* (Figure 5c,d). Whether or not phytoceramides may be helpful for reducing cisplatin-induced toxicity in combination therapy, depends on the further experiments with these compounds. On the other hand, phytoceramides can be not only cytoprotective, but also cytotoxic agents for some tumor cells. The “dual” properties of the phytoceramides should be taken into account in an evaluation of their antitumor potential.

Using an in vitro paraquat model of Parkinson’s disease, we found that the phytoceramides from *M. clathrata* influenced the neurodegenerative effect induced by paraquat. After a short time of pre-incubation (1 h), these phytoceramides decreased ROS formation and potentiated the neurodegenerative effect of paraquat. This looks contradictory because paraquat is known to exert deleterious effects through oxidative stress ([37] and references cited therein). Thus, the reason for the observed additive damaging effect of phytoceramides and paraquat was unclear. However, we admitted that the absorption of phytoceramides by cells during 1h pre-incubation was insufficient to cause the cytoprotective properties of these compounds, and longer periods of time were required for this, as in the cases with crambescidin 359 and cisplatin (Figure 4 and Figure 5). Indeed, the neuroprotective effect of phytoceramides was then revealed in the result of their 24 and 48 h pre-incubation with neuroblastoma cells (Figure 8). This suggests that phytoceramides, reducing neurodegeneration caused by paraquat, may be potential prophylactic agents for decreasing the risk of Parkinson’s disease.

In general, the long-term preliminary treatment of the cells with the phytoceramides of *M. clathrata* was necessary for their cytoprotective functions; otherwise, an additive damaging effect of these sphingolipids and cytotoxic compound (crambescidin 359, cisplatin or paraquat) was observed. Therefore, exogenous phytoceramides can exert dual effects on cell survival, but their “delayed” effect may be cytoprotective.

## Data Availability

Not applicable.

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
