# Peer review of "Phytoceramides from the Marine Sponge Monanchora clathrata: Structural Analysis and Cytoprotective Effects"

_biomolecules, 2023, doi:10.3390/biom13040677_

Round 1

Reviewer 1 Report

The manuscript entitled "Phytoceramides from the marine sponge Monanchora clathrata: Structural Analysis and Cytoprotective Effects" authored by Elena A. Santalova et al. presents interesting data on phytoceramides characterization and cytoprotective potential

The following modifications are recommended:

1.Reformulate the abstract to point out the novelty of the study (first study) and to include conclusions 

2. Reformulate the introduction to avoid word repetitions (ceramides/ceramid - lines 31-38), and to fit the manuscript structure - aim and novelty, while the parts that present results and conclusions (eg lines 75-79) should be inserted in abstract and conclusions sections 

3. Ethics Statement should be added for the use of sponge Monanchora clathrata

Reviewer 2 Report

It is a generally well-written manuscript, where the isolation and activity of phytoceramides is studied. Specifically, it indicates that 16 new (1b, 3a, 15

3c, 3d, 3f, 3g, 5c, 5d, 5f, 5g, 6b–g) compounds were elucidated. These new compounds are determined based on the fragmentation spectrum obtained in mass spectrometry. I would like to know if nuclear magnetic resonance is available for each of the new compounds mentioned above, since they do not appear in the supporting information. This would be a very solid analytical foundation for the manuscript.

Reviewer 3 Report

This manuscript describes the comprehensive analysis of phytoceramides from the marine sponge Monanchora clathrate by structural characterization and bioactivity evaluation. The authors report 16 new and 12 known compounds. Their structures were impressively determined by applying a combination of chemical and instrumental methods including RP-HPLC, NMR, ESI-MS, ESI-MS/MS, and GC-MS. In addition, the authors showed that the cytotoxic effect of crambescidin 359 and cisplatin decreased after pre-incubation of MDA-MB-231 and HL-60 cells with the investigated phytoceramides. Also, phytoceramides decreased neurodegenerative effect and ROS (reactive oxygen species) formation induced by paraquat in neuroblastoma cells. In general, this is an excellent report of phytoceramides from sponges and deserve a publication in Biomolecules.

The only concern is that the authors should discuss what is reason for structure differences of phytoceramides isolated from the same the sponge M. clathrate of Philippine sample and the Australia waters.
